# Development and Psychometric Evaluation of the Chinese Version of the Life Skills Scale for Physical Education

**DOI:** 10.3390/ijerph19095324

**Published:** 2022-04-27

**Authors:** Xiangbo Ji, Shaofeng Zheng, Chuanyin Cheng, Liping Cheng, Lorcan Cronin

**Affiliations:** 1School Physical Educational and Sport Science, Nanjing Normal University, Nanjing 210000, China; jxb18750774277@163.com; 2Sports Department, The Open University of Fujian, Fuzhou 350103, China; zhengshaofeng0119@163.com; 3School Physical Education and Sport Science, Fujian Normal University, Fuzhou 350108, China; 4Department of Sport & Physical Activity, Edge Hill University, Ormskirk L39 4QP, UK; lorcan.cronin@edgehill.ac.uk

**Keywords:** life skills, physical education, positive youth development, translation, validation

## Abstract

Research on life skills in physical education (PE) has gained great attention in recent years. However, there is a need to translate life skills measures for PE into other languages. This research adapted the Life Skills Scale for PE (LSSPE) into Chinese and provided evidence for its validity and reliability. In Study 1, the scale was cross-culturally adapted through translation and back-translation, expert feedback, pilot testing, and scale refinement to provide evidence for the content validity of the scale. Study 2 provided evidence for the factorial validity, internal consistency reliability, and test–retest reliability of the scale by testing it with 583 students. Study 3, with 390 students, provided evidence for the nomological validity of the measure, with results showing perceived teacher autonomy support and students’ basic need satisfaction were positively associated with life skills development in PE, and that controlling teaching and basic need frustration were negatively associated with life skills development. In conclusion, the results illustrate that the LSSPE can be used to evaluate Chinese-speaking students’ life skills development in PE.

## 1. Introduction

Life skills are defined as the skills that are required to deal with the demands and challenges of everyday life [1]. Examples of life skills include leadership, goal setting, social skills, and emotional skills. These life skills can be utilized within different areas of a person’s life (e.g., education and work). Benson [2] proposed the “pile-up” effect, which describes that greater overall life skills are correlated with better life outcomes in a variety of areas. More specifically, Bailey et al. [3] proposed that life skills can enhance young people’s educational achievements, quality of life, and future economic prosperity.

Two important settings in which young people develop life skills are youth sports and physical education (PE). Life skills development in sports has been extensively researched [4,5,6,7], whereas less research has focused on life skills development in PE. Nonetheless, a key aim of PE is the personal development of students [8], which includes the life skills that young people learn through engaging in PE [9]. Past research has shown PE to be a viable and promising context for developing students’ life skills [9,10,11,12,13,14]. For example, Pesce et al. [11] discovered that implementing a life skills program in PE promoted the development of students’ cooperation, goal setting, and decision making.

Four primary reasons may account for why students develop these life skills specifically through PE. First, it is likely that the interactive (i.e., working with more than one person), social (i.e., socializing with different people), and emotional (i.e., dealing emotionally with either success or failure) nature of PE give students’ ample opportunities to develop these skills [15,16,17]. Second, PE classes are typically required in school and can therefore reach most students. Third, students are guided by trained teachers [18] who are responsible for creating pedagogical circumstances under which positive outcomes should result [19]. Finally, studies have shown that varied teaching methods can promote the development of different life skills. For example, cooperative learning in PE helps students develop their teamwork, communication, and leadership skills [20]. Nonetheless, when compared to sports, less is known about the mechanisms through which students develop these life skills [13,21]. Future research on the contribution of PE to life skills development should use valid and reliable instruments to measure the extent to which students develop life skills specifically through PE.

One measure that has been used to assess life skills development in PE is the Life Skills Scale for PE (LSSPE) [13], which was a modified version of the Life Skills Scale for Sport (LSSS) [4]. Based on the eight most-frequently cited life skills within the literature [22], Cronin and Allen [4] undertook a research program to develop and validate the LSSS. This 43-item scale evaluates the following life skills: teamwork, goal setting, social skills, problem solving and decision making, emotional skills, leadership, time management, and interpersonal communication skills. Cronin et al. [21] then modified the scale for use within PE, creating the LSSPE. The difference between the LSSS and LSSPE is the item stem, where the LSSS item stem is “This sport has taught me to…”, while the LSSPE uses the stem “PE classes have taught me to…” Cronin et al. [13,14,21] supported the LSSPE by providing evidence for its factorial validity, nomological validity, and internal consistency reliability. The LSSPE serves as a useful tool for researchers to explain the theories and mechanisms for developing life skills in PE. Past studies using the LSSPE [13,14,21], have utilized the measure to investigate how self-determination theory [23] may be applied to life skill development in PE. The first of these studies showed that teacher autonomy support—which refers to the ways in which teachers help students understand the purpose of their activities and gain the most benefit from them—is associated with students’ total life skills development in PE, and this is in turn associated with students’ psychological wellbeing [21]. The second study showed that autonomy, competence, and relatedness satisfaction are important mechanisms that explain the relationships between perceived teacher autonomy support and student’s life skills development [13]. The third study which was longitudinal in nature showed that total need satisfaction measured at time 1 predicted students’ life skills development at time 2 [14]. Overall, the LSSPE has helped elucidate some of the causes and effects of life skills development amongst English-speaking PE students.

The LSSS has been translated into several languages thus far, including Portuguese [7], French [24], Korean [25], and Turkish [26]. However, there is no scale specifically designed to measure life skills development through PE in languages other than English. Based on the similarity between the LSSS and the LSSPE, it is reasonable to speculate that the LSSPE may be adapted and used to assess the development of life skills in non-English speaking PE students.

Clearly, the need to explore life skills development in PE should not be restricted to only English-speaking students; thus, there is a need to develop a valid and reliable scale in other languages. Specifically, the lack of effective tools to study the life skills of individuals in other countries, cultures, and contexts may hinder a broader understanding of life skills development across the world. In this regard, Moustaka et al. [27] indicated that when applying theories and models in different countries and cultures, it is important to develop linguistically and culturally appropriate measures. Thus, for scales such as the LSSPE to operate in other languages, they should be adjusted and validated in various other languages. The most widely spoken language in the world, Chinese, would serve as a useful language to develop and test the LSSPE. Chinese is one of the six working languages of the United Nations and has more than 1.5 billion speakers in China, Hong Kong Special Administrative Region, Macau Special Administrative Region, Singapore, Malaysia, and Thailand, among other regions. To date, no research has been conducted on the development of life skills among Chinese PE students. More surprisingly, Opstoel et al. [19] reviewed 88 articles on personal and social development in PE and sports from 2008 to 2017 and found that more than half of the studies took place in North America, a quarter were conducted in Europe, and none involved China. Thus, the development of a Chinese version of the LSSPE can help address Opstoel et al.’s [19] proposition that this research should be extended to other continents, countries, and cultures.

In China, PE is a compulsory subject for all students from the first grade of primary school to grade two of university and it is included as an exam subject in the majority of schools. Students typically focus on learning motor skills and engaging in exercise and physical activity. Additionally, the core purpose of an ongoing educational revolution in China is promoting the overall development and healthy growth of primary and secondary school students. Under this educational reform, PE classes will undergo unprecedented development, meaning that the LSSPE has the potential to present rich opportunities for the improvement of PE classes in China.

Specifically, the development of a Chinese version of the LSSPE would allow for future studies on life skills development in PE among Chinese speakers. Examples of potential contributions of the scale are to: (a) assess life skill programs currently or potentially used in Chinese-speaking regions of the world, (b) research the antecedents and consequences of life skills development (e.g., investigate how the PE teaching climate and students’ wellbeing is associated with life skills development), (c) test different theories and models of development (e.g., self-determination theory) with the life skills scale as an outcome variables, and (d) help advance Chinese PE teaching practices (e.g., assess the effectiveness of potential implicit and explicit strategies used to teach life skills in PE).

Accordingly, the overall purpose of this research was to translate the LSSPE [21] into Chinese, as well as to assess the validity and reliability of the newly translated scale. Three aspects of The Standards for Educational and Psychological Testing (The Standards) [28] were assessed: test content, internal structure, and relationships to other variables. More specifically, across three studies, we assessed the content, factorial, and nomological validity of the scale, along with the internal consistency reliability and test–retest reliability of the scale.

## 2. Study 1: Test Content

The purpose of Study 1 was to translate and adapt the LSSPE into Chinese. Accurate translation and adaptation of the measure are necessary to ensure that measurement tools can successfully adapt to the target language and be used properly in the sociocultural context. Following Nascimento Junior et al.’s [7] procedure for translating the LSSS into Portuguese, the relevance, representativeness, and technical quality of the scale items are all important for maintaining validity [7]. Content validity refers to whether items in a measure are “relevant to and representative of the targeted construct for a particular assessment purpose” [29] (p. 239). Subject experts can also be employed to evaluate the accuracy of translation to ensure that each item provides adequate description to retain its original purpose [7,30].

### 2.1. Methods and Materials

#### 2.1.1. Participants

After getting the permission of the original author, a group of ten experts was assembled to oversee the process of translating and adapting the LSSPE from English to Chinese. The group included six academics with previous experience in scale development and four scientific translators. These six academics were full-time university professors; three work in the field of sports psychology, two work in physical education and sports science, and one specialized in measurement and evaluation. The four translators were experienced in translating scientific texts, and all received academic training in English-speaking countries. All of the experts were Chinese, spoke Chinese as their first language, and were proficient in English as a second language. In the first study, 30 Chinese students between 12–21 years old—a sample of students representing future use of the scale—were invited to pilot test the scale and provide a preliminary assessment of their understanding of each item and the overall content of the Chinese LSSPE (C-LSSPE).

#### 2.1.2. Procedure

The first step in assuring content validity was to accurately translate the LSSPE through expert translation from English to Chinese. Following procedures used by Nascimento Junior et al. [7] to translate the LSSS to Portuguese, using a five-point Likert scale (1 = not at all, 5 = very much), experts were asked to rate the clarity of each item, its relevance for inclusion, and to organize each item by determining the life skill it assesses. This process helped ensure the relevance, representativeness, and technical quality of the items and has been used in other scale adaptation and translation studies [7,31]. The original Chinese translation was followed by a back-translation to English to ensure its accuracy. This allowed researchers to obtain two English versions of each questionnaire (e.g., the original version and the retranslated version) and helped ensure the accuracy of the translation. After this, the expert committee worked together to discuss and refine the Chinese translation and English back-translation following procedures described by Ciampolini et al. [32]. Any unclear and inaccurate expressions were modified until all members of the expert team reached a consensus. Alterations were typically made to conform to standards in Chinese items, such as changing “accept suggestions for improvement from others” to “be able to accept suggestions from others to improve yourself”.

The updated version of the C-LSSPE was then pilot-tested among 30 Chinese PE students aged 13–21 years old (15 males and 15 females) to assess the clarity of the questions. All students provided informed consent to participate in the study. They were all students who participated in PE and sports activities. By investigating students’ comprehension and understanding of the scale among a small sample size, the appropriateness of the scale with the target group can be predicted. Other scale validation studies have used this methodology [7,32,33]. After students completed the scale, they were each briefly interviewed by the primary researcher about their perceptions of the items, and were given an opportunity to suggest improvements. The items that students were unsure of and their suggestions for improvement were documented during this process. Subsequently, the expert group met on one last occasion to discuss the students’ feedback and suggestions, which they used to make minor modifications to the C-LSSPE. Specifically, they changed the wording of some items in Chinese. For example, the social skill item “start a conversation” was changed to “initiate a conversation”, and the problem-solving item “think carefully about a problem” was changed to “be able to think carefully about a problem”.

#### 2.1.3. Content Validity Data Analyses

A content validity assessment was performed by analyzing the 43 items of the C-LSSPE to check expert agreement about categorizing items by particular life skill [34]. Consistent with previous studies [7,35], we calculated the coefficient of content validity (CCVi) for each item and for the scale as a whole (CCVt) to assess the language clarity and practical relevance of items. To analyze the agreement of experts, a kappa coefficient value of 0.80 or above was deemed acceptable [36].

### 2.2. Results

The aim of Study 1 was to create a Chinese version of the LSSPE. The content validity assessment demonstrated that the life skills items in the C-LSSPE used clear language and demonstrated practical theoretical relevance, with coefficients of content validity above 0.80 (CCVi ranged from 0.92–1.0, *M* = 0.94; CCVt ranged from 0.86–1.0, *M =* 0.90). These findings suggested that the C-LSSPE presents clear language and relevance for Chinese-speaking students, and it maintains its validity in the context of PE. The C-LSSPE item classification agreement among experts (Kappa coefficient) for teamwork, goal setting, social skills, time management, problem solving and decision making, emotional skills, leadership, and interpersonal communication skills ranged from 0.85 to 0.91 (*M* = 0.87), indicating that items corresponded to their correct underlying dimension. After completion of the C-LSSPE translation steps, statistical findings indicated that translations and feedback from experts and students had produced a scale with adequate content validity.

## 3. Study 2: Internal Structure

The aim of Study 2 was to assess the internal structure of the C-LSSPE. Specifically, we used confirmatory factor analysis (CFA) to assess the factorial validity of the scale using scores obtained from a large sample of Chinese PE students. Factorial validity involves measuring items correlating strongly with their theoretical construct (i.e., a teamwork item correlates strongly with the teamwork latent variable) while correlating weakly or not at all with other theoretical constructs [37]. During this study, we also tested the internal consistency reliability and test–retest reliability of the C-LSSPE subscales and total life skills score. Internal consistency reliability refers to the extent to which each item in a scale or subscale is measuring the same variable [38], whereas test–retest reliability refers to the stability of scores over time [39].

### 3.1. Methods and Materials

#### 3.1.1. Recruitment

The first author contacted the PE department at the students’ schools to obtain approval for data collection for this study. Each student also provided informed consent before taking part in the study. Our inclusion criteria for the study were middle school, high school, and first-year college students who normally participate in PE classes and sports activities. In China, the age range of students in these schools is usually 13 to 15 years old for middle school students, 16 to 18 years old for high school students, and 19 to 22 years old for college students. It is worth noting that there are several special circumstances that may create a mismatch between the age of a student and the period of study, such as starting school late or repeating a school year. Data collection was conducted in the playground or in classrooms where students were attending classes, at a time agreed upon by the researchers, PE departments, and participants.

#### 3.1.2. Measure

The 43-item C-LSSPE was used to assess how participants perceived the development of the eight life skills in PE. The item stem for this scale was “PE classes have taught me to…” and example items included: teamwork (seven items; e.g., “work well in a team/small group”, goal setting (seven items; e.g., “set goals so I can focus on improving”), social skills (five items; e.g., “initiate a conversation”), problem solving and decision making (four items; e.g., “be able to think carefully about a problem”), emotional skills (four items, e.g., “know how to deal with emotions”), leadership (eight items; e.g., “know how to positively influence a group of people”), time management (four items; e.g., “reasonable arrangement of time”), and interpersonal communication (four items; e.g., “talking clearly to others”). Students responded to the items on a five-point Likert scale (1 = not at all, 5 = very much). The complete C-LSSPE is available in Appendix A.

#### 3.1.3. Participants

A total of 630 students (304 males and 326 females) from all regions of China participated in this study. However, 47 students were excluded from the final sample because they did not adequately or clearly respond to the survey (i.e., they failed to respond to items or they responded more than once to the same item). In total, 583 students (290 males and 293 females) aged between 13–21 years (*M*_age_ = 15.51 years; SD = 2.55 years) were in the final sample. The students took part in PE for an average of 115.78 min per week (SD = 22.26 min). In PE classes, the students participated in a wide range of sports, including basketball, football, volleyball, track and field, martial arts, and tai chi. They participated in extracurricular sports at school for an average of 95.63 min per week. Furthermore, 40.14% of students took part in sports outside of school for an average of 150 min per week. From the sample population, 127 students (70 males and 57 females) aged between 14–21 years old (*M* = 16.76; SD = 2.28 years) were randomly included in a retest of the measure within two weeks of first completing the scale.

#### 3.1.4. Data Analyses

To assess the factorial validity of the C-LSSPE, CFA employing robust maximum likelihood estimation was conducted using AMOS software (Version 23.0, IBM Corporation, NY, the United States of America) [40]. This involved testing an eight-factor model representing all eight life skills, as well as a second-order model that included all eight life skills and a total life skills factor. The goodness of fit of the scale was evaluated by four fit indexes: chi-square divided by degrees of freedom (χ^2^/df), root mean square error of approximation (RMSEA) [41], comparative fit index (CFI) [42], and the Tucker–Lewis index (TLI) [43]. A χ^2^/df of less than 3.0 was indicative of adequate fit [44]. A RMSEA value of less than 0.08 or 0.05 represented a reasonable or close fit to the data, respectively. CFI and TLI values greater than 0.90 or 0.95 indicated acceptable and excellent fit, respectively [45]. Factor loadings were judged according to Comrey and Lee’s [46] criteria of loadings greater than 0.71 considered excellent, 0.63 as very good, 0.55 as good, 0.45 as fair, and 0.32 as poor.

The internal consistency reliability of the C-LSSPE was assessed through the Cronbach’s alpha, with values above 0.70 indicating acceptable internal consistency [36]. Furthermore, the test–retest reliability of the scale was assessed using intraclass correlation coefficients (ICCs). ICC values were judged as poor if they were 0.20 or less, reasonable from 0.21 to 0.40, good from 0.41 to 0.60, very good from 0.61 to 0.80, and excellent from 0.81 to 1.00 [47].

### 3.2. Results

#### Factorial Validity Assessment

Table 1 includes the fit indices for the two models tested. The first-order model and second-order model both displayed an acceptable fit. Figure 1 and Figure 2 include the factor loadings for the two models tested. In the first order model, the factor loadings ranged from 0.60 to 0.85 (*M* = 0.73). Among them, 30 items had factor loadings that were considered excellent and 11 items had factor loadings that were considered very good. Only two items had factor loadings less than 0.63, which was still designated as good. Within the second-order model, all life skills factors loaded significantly onto the higher order factor, and the factor loading coefficients ranged from 0.70 to 0.91 (*M* = 0.81). This indicated that all eight subscales of the C-LSSPE can be combined to calculate a total life skill scale. Table 2 shows the correlation between the eight life skills, with values ranging from 0.47 to 0.78. Importantly, none of the correlations were greater than the 0.80 used to identify poor discriminant validity [48].

### 3.3. Internal Consistency Reliability

Table 3 shows that the Cronbach’s alpha coefficients of the subscales and total life skills score ranged from 0.80 to 0.96. These values were all above the 0.70 recommended by Nunnally and Bernstein [36] for adequate internal consistency reliability. In terms of test–retest reliability, the ICC values were above 0.63 (*M* = 0.65), apart from the emotional skills subscale (ICC = 0.54). According to Weir [47], these ICCs were considered good or very good, thus supporting the test–retest reliability of the scale.

## 4. Study 3: Relationships to Other Variables

The aim of this study was to test whether the C-LSSPE scores correlated with theoretically relevant outcomes, in order to test nomological validity. Nomological validity refers to a construct’s relationship with other theorized constructs [49]. Self-determination theory (SDT) [23] has been widely used to investigate coaching and teaching behaviors and young people’s life skills in sports and PE [4,13,14,50]. According to SDT, people will develop positively if certain environmental conditions are present [51]. One aspect of SDT is a teacher’s interpersonal style, which can be conceptualized in terms of autonomy-supportive and controlling teaching [52]. Another aspect of SDT is the degree to which PE students’ psychological needs for autonomy, competence, and relatedness are satisfied or frustrated [53]. Several studies have highlighted that teacher autonomy support and satisfaction of students’ three basic needs are positively related to life skill development [13,14,21]. Researchers have also highlighted that controlling teaching and frustration of these three basic needs have negative effects on students’ adaptive outcomes in PE [53,54,55]. Thus, in the current study, it was hypothesized that teacher autonomy support and satisfaction of students’ three basic needs would positively relate to students’ life skills development in PE, while controlling teaching and frustration of students’ three basic needs would be negatively associated with life skills development.

### 4.1. Methods and Materials

#### 4.1.1. Participants

The sample included 390 students (215 males and 175 females) between 12–21 years old (*M*_age_ = 16.32; SD = 2.93). The students were recruited individually, were enrolled in school, and participated in PE and sports activities. The students took part in PE for an average of 112.15 min per week (SD = 25.73) and were taking PE as an exam subject. In PE classes, the students participated in a range of sports, including basketball, football, volleyball, track and field, martial arts, and tai chi. They participated in extracurricular sports at school for an average of 112.96 min per week. Furthermore, 38.97% of students took part in sports outside school, for an average of 105.59 min per week.

#### 4.1.2. Measures

##### Life Skills Development

The 43-item C-LSSPE described in Study 2 was used to measure students’ perceived life skills development. With the current sample, the C-LSSPE subscales displayed Cronbach’s alpha coefficients ranging from 0.82–0.97 (Table 4), which supported the internal consistency reliability of the subscales.

##### Autonomy-Supportive and Controlling Teaching

The six-item Chinese version of the Health Care Climate Questionnaire (HCCQ) [56] was used to measure students’ perceptions of teacher autonomy support. The six-item Chinese version of the Psychological Control in Teaching Scale (C-PCTS) [57] was used to measure students’ perceptions of controlling teaching behaviors. Participants responded to the items using a seven-point Likert scale (1 = strongly disagree, 7 = strongly agree). The C-HCCQ and C-PCTS have been used with Chinese-speaking students and demonstrated good reliability in previous studies (e.g., C-HCCQ, composite reliability = 0.84; C-PCTS, composite reliability = 0.78) [54]. In the current study, the C-HCCQ and C-PCTS displayed Cronbach’s alpha coefficients of 0.93 and 0.83 respectively, indicating adequate internal consistency reliability [36].

##### Basic Needs Satisfaction and Frustration

The Chinese version of the psychological needs satisfaction scale for PE (PNSSPE) [58] and psychological needs frustration scale for PE (PNTSPE) [59] were used to assess students’ need satisfaction and frustration. The item stem for the PNSSPE and PNTSPE was “In my physical education classes, I…”. The PNSSPE includes ten items that measure three aspects of satisfaction with: autonomy (e.g., I have opportunities to express my views and thoughts), competence (e.g., I get opportunities to feel that I am good at sports), and relatedness (e.g., I feel comfortable when being with people). The PNTSPE includes nine items that measure frustration of: autonomy (e.g., I feel pushed to behave in certain ways), competence (i.e., I often feel incompetent), and relatedness (e.g., I feel I am rejected by those around me). Participants responded to the items using a seven-point Likert scale (1 = strongly disagree, 7 = strongly agree). The subscales demonstrated good internal consistency reliability in previous studies (PNSSPE; three basic needs satisfaction subscales ranged from 0.81–0.85; PNTSPE; three basic need frustration subscales ranged from 0.79–0.84) [54,58,59]. In the current study, the Cronbach’s alpha coefficients for the need satisfaction subscales ranged from 0.85–0.93, and for the three need frustration subscales they ranged from 0.87–0.92.

#### 4.1.3. Data Analysis

Correlation coefficients were used to assess the relationships between teacher autonomy support, students’ basic psychological need satisfaction, and life skills development. A *p*-value of less than 0.05 was required to indicate a statistically significant relationship between variables. The correlations were judged as small (*r* = ±0.10 to ±0.29), medium (*r* = ±0.30 to ±0.49), or large (*r* > ±0.50), based on Cohen’s criteria [60].

### 4.2. Results

Table 4 shows that teachers’ autonomy-supportive behaviors and satisfaction of the three basic needs displayed significant positive correlations with each of the eight life skills and total life skills. These correlations ranged from 0.32 to 0.47 and could be judged as medium in size [60]. Conversely, the teachers’ controlling behavior and frustration of the three basic needs displayed significant negative correlations with the eight life skills and total life skills. These correlations ranged from −0.16 to −0.37. These negative correlations were small in size, apart from the medium-sized negative correlations between competence frustration and leadership and total life skills, and between relatedness frustration and social skills, emotional skills, and total life skills. Notably, there was a small negative correlation between total need frustration and goal setting, time management, and communication skills; and a medium-sized negative correlation with the other life skills. Overall, the results suggested that teacher autonomy support and students’ basic need satisfaction can promote the development of students’ life skills, while controlling teaching behaviors and basic need frustration inhibit the development of students’ life skills.

## 5. Discussion

The purpose of this research was to adapt the LSSPE to Chinese and provide validity (e.g., test content, internal structure, and relationship to other variables) and reliability evidence (e.g., test–retest reliability and internal consistency reliability) for the scale. This was the first attempt at adapting scales that assess life skill development to Chinese that we are aware of. As with previous sports studies [7,24], this study further expands the assessment of life skills development to a new language. Chinese, the language with the largest number of speakers worldwide, is a viable target for translation of the LSSPE. The C-LSSPE can be used by PE teachers, researchers, and practitioners to assess Chinese students’ life skills development in PE and investigate any antecedent variables that may impact life skills development (e.g., PE teaching behaviors) or subsequent variables that may be impacted by life skills development (e.g., students’ psychological well-being). Furthermore, different theories could be tested in PE using the scale as an outcome variable (e.g., SDT or achievement goal theory).

In Study 1, we adapted and translated the LSSPE into Chinese and provided evidence for the content validity of the scale items. First, we utilized content experts—academics and translators—and a sample of students to ensure that the content validity of the original scale was maintained when translated into Chinese. We also provided evidence for the validity of the scale by calculating and presenting CCVi and CCVt values to support the content validity of the scale, which is an important and often neglected factor during the process of scale development in sports psychology [61].

Study 2 provided evidence for the validity of the internal structure of the C-LSSPE with a large sample of students. The factorial validity of a first-order model and a second-order model was supported. Additionally, each life skill subscale and total life skills showed acceptable internal consistency reliability and test–retest reliability. Thus, the C-LSSPE can be used reliably with Chinese students to investigate the development of each of the eight life skills separately and total life skills combined. These findings align with the factorial validity and internal consistency reliability evidence for the LSSPE with English-speaking students [13,14,21]. Importantly, this scale helps to break through the linguistic limitations of studying life skills development in PE and ensures that research in this area is not limited to English-speaking countries.

Study 3 provided evidence for the relationships with other variables of the C-LSSPE by illustrating the expected pattern of relationships between teaching behaviors, basic need satisfaction and frustration, and life skills development. These findings support earlier research showing that PE teacher autonomy support is positively associated with students’ life skills development [13,21]. This is an important finding because it shows that life skills in Chinese-speaking students can also be developed through teachers implementing more autonomy-supportive behaviors. Like previous studies, satisfaction of students’ three basic psychological needs was positively related to adaptive student outcomes in PE [62], including life skills development [13,14]. Notably, we also found that teachers’ controlling behaviors and the frustration of competence, autonomy, and relatedness had significant negative associations with students’ life skills development. Though the correlations between most of these variables were generally small in size, this result implies that controlling teaching behaviors and frustration of students’ psychological needs will inhibit the development of students’ life skills. Interestingly, this finding differs from Cronin et al.’s [13] study, which found no statistically significant relationships between students’ perceptions of controlling teaching and PE students’ life skills development. This may be due to cultural differences between British and Chinese students’ perceptions of their PE teachers and the PE environment. Like the present study, several other researchers have also found that students’ perceptions of controlling teaching have negative associations with PE students’ engagement [63], autonomous motivation [53], subjective vitality [54], and prosocial behavior [55]. Based on findings in this study, teachers should display more autonomy-supportive behaviors in PE classes, such as providing choice in activities, encouraging students to work together and independently, acknowledging students’ feelings, and providing a rationale for class activities. Additionally, teachers should aim to create an environment where students’ three basic needs are satisfied.

## 6. Limitations and Future Research Recommendations

Despite its contributions, our work has some limitations. First, in the present study, all data were collected via student self-report, which can be affected by both social desirability and memory recall [64]. Thus, future research could measure coach autonomy support and controlling teaching using observational methods and measure life skills via teacher/parent ratings [13]. Second, students participating in this study are from mainland China. Considering that there are variations in PE teaching in different regions of China, as well as different PE teaching policies in other Chinese-speaking countries, future research should include a wider range of Chinese-speaking cities or countries to further the validation process. Third, the students in our study were aged 12–21 years old; thus, the applicability of this scale to students of different gender or age groups requires further validation. For example, future studies should include a larger sample size to verify the invariance of the measure across older versus younger age groups (e.g., 11–14 years old and 15–18 years old) and genders. Finally, a limitation of the current research is that only certain elements of validity were tested across the three studies. Future studies should look to assess additional aspects of validity, such as predictive validity (e.g., to assess whether life skills development is associated with other positive outcomes such as psychological well-being).

## 7. Practical Implications

Overall, our development of the C-LSSPE extends the research on life skills development in PE from the English-speaking to the Chinese-speaking context. To begin with, the C-LSSPE will allow practitioners and researchers to explore Chinese students’ life skills development in PE. On the one hand, this will help broaden the scope of research on youth development beyond English speakers in North America [65]. On the other hand, the new findings may provide teachers with information and feedback to promote the healthy development of students and help them better use life skills to adapt to Chinese society. Specifically, the use of the C-LSSPE in conjunction with relevant theories (e.g., SDT) can provide educators and policymakers with theory-based evidence, explanations, and guidance to help them design programs to teach life skills to students. Furthermore, researchers will be able to explore whether Chinese-speaking student’s life skills development in PE differ from those of their English-speaking counterparts. More importantly, given that this was the first research project that used the LSSPE with Chinese-speaking students, researchers can use this to investigate theories (e.g., SDT and achievement goal theory) that that may help explain how students develop life skills in PE.

## 8. Conclusions

Translation of the LSSPE into Chinese and subsequent analyses have demonstrated sufficient evidence for its test content, internal structure, and relationship to other variables (The Standards) [28]. The adapted C-LSSPE can be used to assess the perceptions of students from Chinese-speaking countries concerning their life skills development in PE. Furthermore, this research found that students’ perceptions of teacher autonomy support and basic need satisfaction was positively related to the development of all eight life skills and total life skills. In practice, creating an autonomy-supportive climate and supporting the satisfaction of students’ basic needs is one important way to foster students’ life skills. Thus, based on the findings in this study, teachers should show more autonomy-supportive behaviors in PE classes, such as providing choice in activities, encouraging students to work together and independently, acknowledging students’ feelings, and providing a rationale for class activities. Additionally, teachers should aim to create an environment where students’ three basic needs are satisfied. Finally, we hope our study inspires more research on this issue outside of English-speaking countries, not only for the pursuit of knowledge but also for the promotion of students’ life skills in practice.

## Figures and Tables

**Figure 1 ijerph-19-05324-f001:**
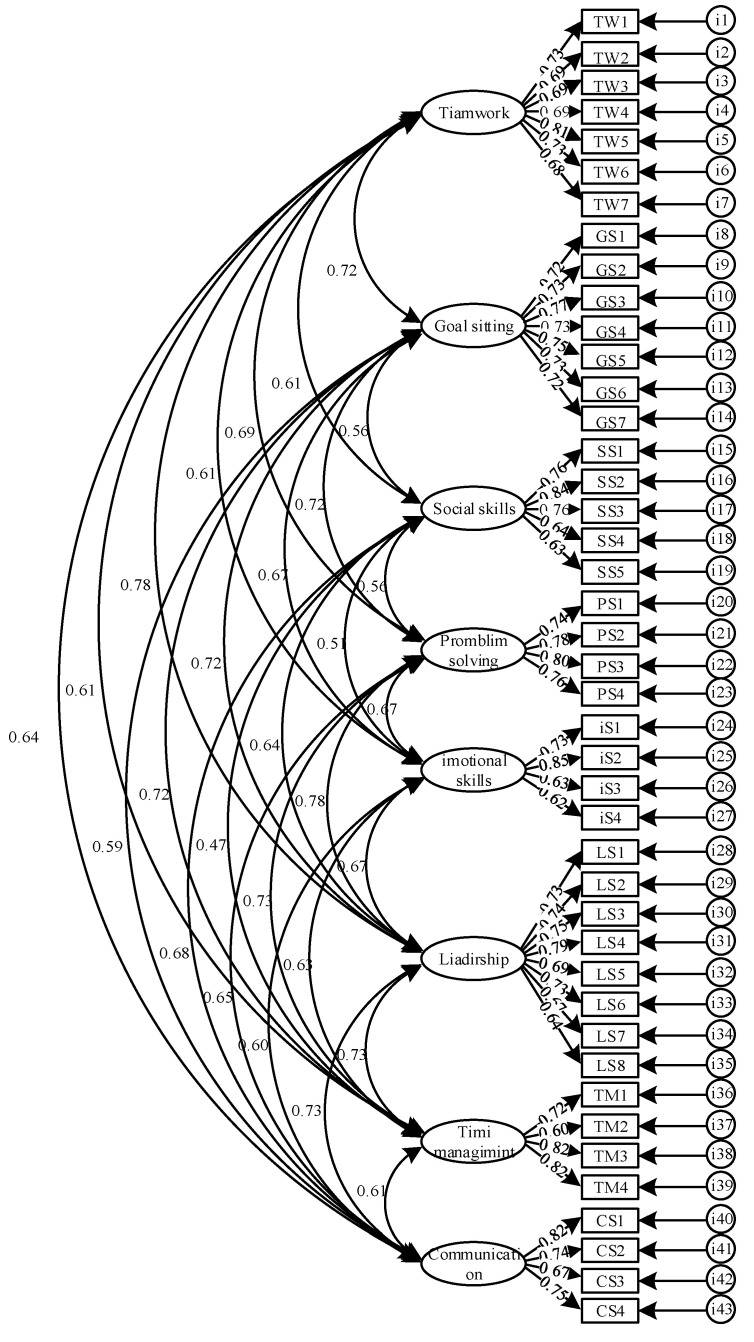
Standardized coefficients for the first-order model.

**Figure 2 ijerph-19-05324-f002:**
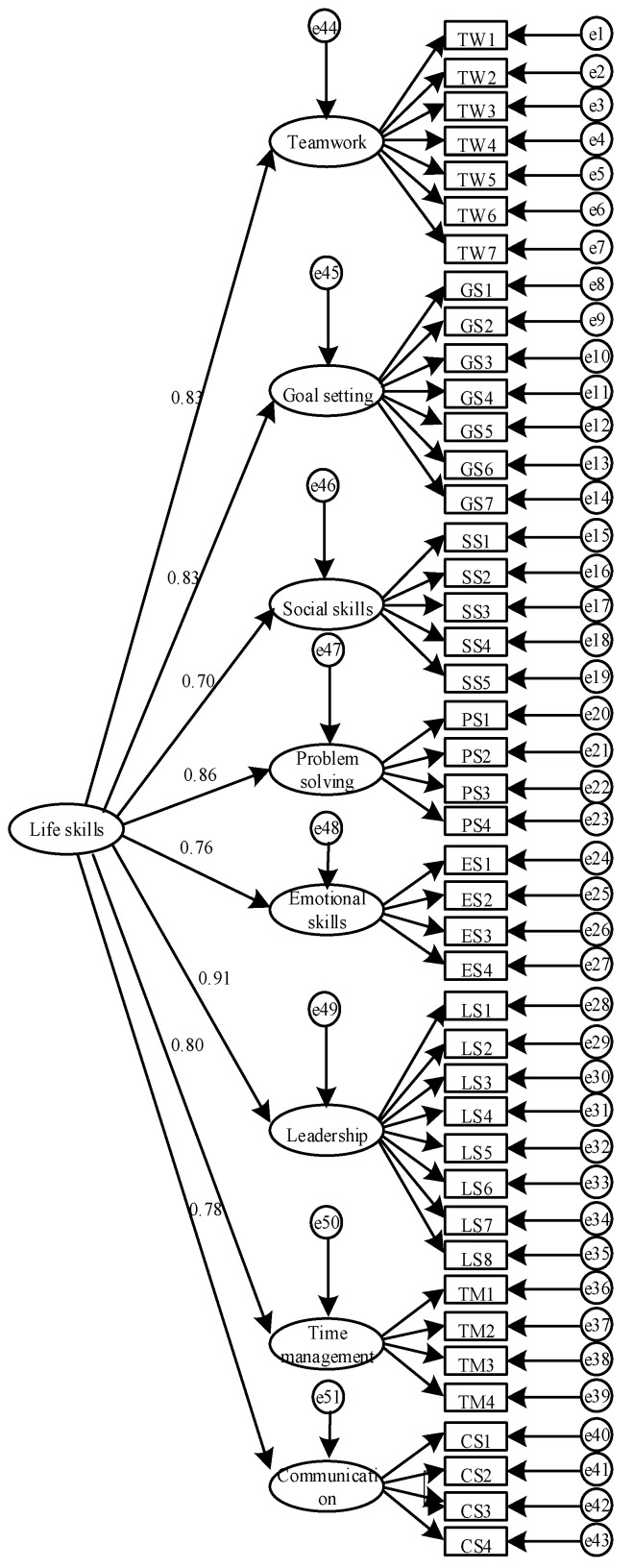
Standardized coefficients for the second-order model.

**Table 1 ijerph-19-05324-t001:** Indices of model fit for the C-LSSPE.

Model	χ^2^	df	χ ^2^/df	RMSEA	CFI	TLI
First-order model (eight life skills)	1845.222 **	832	2.218	0.046	0.927	0.921
Second-order model (eight life skills and a total life skill)	1942.372 **	852	2.280	0.047	0.921	0.917

Note. *n* = 583. RMSEA = root mean square error of approximation; CFI = comparative fit index; TLI *=* Tucker–Lewis index. ** *p* < 0.001.

**Table 2 ijerph-19-05324-t002:** Correlations between the eight life skills in Study 2.

	*M*	SD	1	2	3	4	5	6	7	8
Teamwork	3.52	0.74								
Goal setting	3.51	0.74	0.72 **							
Social skills	3.61	0.83	0.61 **	0.57 **						
Problem solving	3.51	0.77	0.69 **	0.72 **	0.56 **					
Emotional skills	3.55	0.79	0.61 **	0.67 **	0.51 **	0.67 **				
Leadership	3.37	0.75	0.78 **	0.72 **	0.64 **	0.78 **	0.67 **			
Time management	3.37	0.77	0.61 **	0.72 **	0.47 **	0.73 **	0.63 **	0.73 **		
Communication	3.85	0.77	0.64 **	0.59 **	0.68 **	0.65 **	0.60 **	0.73 **	0.61 **	

Note. *n =* 583. Problem solving = problem solving and decision making; Communication = interpersonal communication skills; *M* = mean score; SD = standard deviation. ** *p* < 0.01.

**Table 3 ijerph-19-05324-t003:** Cronbach’s alpha values and intraclass correlation coefficients for Study 2.

	Cronbach’s Alpha (*n* = 583)	ICC (*n* = 127)
Teamwork	0.88	0.71
Goal setting	0.89	0.66
Social skills	0.85	0.63
Problem solving and decision making	0.85	0.66
Emotional skills	0.80	0.54
Leadership	0.89	0.69
Time management	0.82	0.68
Communication	0.83	0.64
Total life skills	0.96	0.75

**Table 4 ijerph-19-05324-t004:** Correlations between students’ life skill scores and teaching behaviors and basic need satisfaction and frustration.

	*M*	SD	1	2	3	4	5	6	7	8	9	10	11	12	13	14	15	16	17	18	19
Autonomy support	5.46	1.28	(0.93)																		
Controlling teaching	2.46	1.11	−0.47 **	(0.83)																	
Autonomy satisfaction	5.25	1.29	0.62 **	−0.35 **	(0.86)																
Competence satisfaction	4.96	1.36	0.46 **	−0.18 **	0.58 **	(0.85)															
Relatedness satisfaction	5.68	1.17	0.46 **	−0.26 **	0.51 **	0.49 **	(0.93)														
Autonomy frustration	2.34	1.30	−0.49 **	0.59 **	−0.45 **	−0.26 **	−0.33 **	(0.92)													
Competence frustration	2.87	1.48	−0.33 **	0.39 **	−0.42 **	−0.52 **	−0.36 **	0.52 **	(0.89)												
Relatedness frustration	2.01	1.11	−0.30 **	0.43 **	−0.27 **	−0.35 **	−0.48 **	0.44 **	0.52 **	(0.87)											
Total need satisfaction	5.29	1.06	0.63 **	−0.33 **	0.88 **	0.83 **	0.77 **	−0.43 **	−0.52 **	−0.43 **	(0.90)										
Total need frustration	2.41	1.06	−0.46 **	0.58 **	−0.47 **	−0.47 **	−0.47 **	0.81 **	0.86 **	0.77 **	−0.57 **	(0.89)									
Teamwork	3.63	0.84	0.46 **	−0.24 **	0.43 **	0.36 **	0.46 **	−0.24 **	−0.25 **	−0.25 **	0.50 **	−0.30 **	(0.93)								
Goal setting	3.66	0.88	0.39 **	−0.18 **	0.45 **	0.45 **	0.36 **	−0.23 **	−0.26 **	−0.21 **	0.51 **	−0.29 **	0.68 **	(0.95)							
Social skills	3.75	0.85	0.35 **	−0.20 **	0.34 **	0.37 **	0.47 **	−0.24 **	−0.28 **	−0.37 **	0.46 **	−0.36 **	0.64 **	0.62 **	(0.88)						
Problem solving	3.69	0.88	0.39 **	−0.27 **	0.39 **	0.40 **	0.36 **	−0.26 **	−0.26 **	−0.29 **	0.46 **	−0.33 **	0.66 **	0.74 **	0.64 **	(0.92)					
Emotional skills	3.70	0.86	0.37 **	−0.26 **	0.38 **	0.35 **	0.38 **	−0.26 **	−0.26 **	−0.32 **	0.45 **	−0.34 **	0.61 **	0.63 **	0.65 **	0.73 **	(0.85)				
Leadership	3.55	0.85	0.44 **	−0.27 **	0.43 **	0.46 **	0.39 **	−0.29 **	−0.32 **	−0.29 **	0.52 **	−0.37 **	0.68 **	0.71 **	0.70 **	0.77 **	0.74 **	(0.93)			
Time management	3.50	0.86	0.36 **	−0.16 **	0.41 **	0.44 **	0.32 **	−0.20 **	−0.29 **	−0.22 **	0.47 **	−0.29 **	0.59 **	0.74 **	0.58 **	0.67 **	0.62 **	0.71 **	(0.88)		
Communication	3.87	0.84	0.38 **	−0.20 **	0.34 **	0.36 **	0.40 **	−0.19 **	−0.20 **	−0.33 **	0.44 **	−0.28 **	0.62 **	0.57 **	0.72 **	0.59 **	0.60 **	0.67 **	0.59 **	(0.88)	
Total life skills	3.66	0.72	0.47 **	−0.27 **	0.48 **	0.48 **	0.47 **	−0.29 **	−0.32 **	−0.33 **	0.58 **	−0.38 **	0.84 **	0.86 **	0.82 **	0.86 **	0.82 **	0.90 **	0.81 **	0.78 **	(0.98)

Note. *n* = 390. Problem solving = problem solving and decision making; Communication = interpersonal communication skills; *M* = mean score; SD = standard deviation. Alpha coefficients for each subscale are contained within the parentheses. ** *p* < 0.01.

## Data Availability

Please note that the data are available upon request from the corresponding author.

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
