# Peer review of "Development and Psychometric Evaluation of the Chinese Version of the Life Skills Scale for Physical Education"

_ijerph, 2022, doi:10.3390/ijerph19095324_

Round 1

Reviewer 1 Report

The topic of this paper is very interesting and it should be studied in future educational research and other review or overview.

Introduction

I recommend that you highlight in more detail the novelty of the study in correlation with previous studies.

Participants

I recommend that you detail the gender distribution of the research sample and what the inclusion and exclusion criteria were for the research.

Procedure

I recommend that you add the study period and its staging.

Conclusion

I recommend that you review and expand on the conclusion to highlight the most relevant results of your study.

Please mention in the content of article the fact that the life skills for physical education can influence the quality of life.

Reviewer 2 Report

The work was clear and well presented and the aims and objectives of the research were explained well. The process of translation and the methods used to pilot were explored in depth.

The structure of the methods may have been more concise if   the recruitment and procedures for study 1 and 2 and 3 were combined so that sampling, methods and ethics could be explored in one section and numbers of participants illustrated for each study together. This have cut out some repetition.

Limitations are noted clearly as well as an evaluative discussion as well as the significance of the study.

Reviewer 3 Report

In this manuscript, three-study investigation is presented in which a Chinese version of the Life Skills Scale for Physical Education (LSSPE) is developed. The topic falls squarely within the domain covered by this journal. The strong rationale provided for translating the LSSPE into Chinese, the high quality of the writing, the thorough sequence of studies, and the careful adherence to contemporary measure development practices are desirable features of the manuscript. These positive impressions notwithstanding, I have several extremely minor suggestions about the manuscript in its current form:

  1. Was parental consent obtained in Studies 2 and 3 for participants who were minors?

  1. The apparent causal inference made in the concluding sentence of section 4.2 (i.e., “Overall, the results suggest that teacher autonomy support and students’ basic need satisfaction can promote the development of students’ life skills, while controlling teaching behaviors and basic need frustration inhibit the development of students’ life skills.”) is inappropriate given the nonexperimental/correlational nature of the study. The wording of similar inappropriate conclusions in the Discussion section (i.e., “This is an important finding, because it shows that life skills in Chinese-speaking students can also be developed through teachers’ autonomy supportive behaviors.” and “Though the correlations between most of these variables were generally small in size, this result implies that controlling teaching behaviors and frustration of students’ psychological needs inhibit the development of students’ life skills.”) should also be modified. Likewise, although the applied recommendations at the end of the Discussion (i.e., “Based on findings in this study, teachers should show more autonomy supportive behaviors in PE classes, such as providing choice in activities, encouraging students to work together and independently, acknowledging students’ feelings, and providing a rationale for class activities. Additionally, teachers should aim to create an environment where students’ three basic needs are satisfied.”) are not necessarily unsound, they do not follow directly from the correlational findings of the current study.

  1. In the “Limitations and future research directions section,” it would be appropriate to discuss the lack of criterion-related validity for the LSSPE and recommend future research to address this issue. In particular, the LSSPE relies exclusively on self-report. There is no evidence that higher scores correspond to the actual learning of life skills.

Reviewer 4 Report

The article presents 3 studies in which the Life Skills Scale for Physical Education (LSSPE) is translated, adapted and validated to the Chinese language and context.
This is a suitable theme for the journal.
The article is well written and organized.
The methodology used is rigorous.
The results may be of interest both to researchers interested in using this instrument and to professionals in the practice of physical education interested in evaluating the impact of their classes on their students.

I congratulate the authors for their work.
Here are some suggestions that I hope will help improve the manuscript:

1. It is suggested to divide the headings 3.1.2. and 4.1.1. "Measure and participants" in two different sections: one on the measure and the other on the participants.

2. It is suggested to specify how the 127 students who participated in the retest were chosen.
Was there any specific criteria? Or were they chosen randomly? (My opinion is that they should have been chosen randomly).

3. Likewise, it is suggested to specify if the 390 students who participated in study 3 are a subsample of study 2.
Or if, on the contrary, they are independently recruited participants. In this case, the recruitment method should be briefly explained.

Formal issues.
1. Keywords must be in alphabetic order.
2. References must be written according to the journal style. It's recommended to include a doi and to write the journal name in an abbreviate form.
3. Figures 1 and 2. Erratum: "commiunication".
4. Figure 1. What's the meaning of the letter "e" before each item number? It's suggested to consider to change it by "i" (item).
5. Figure 1. Some numbers are read with difficulty.
6. Table 3. It's suggested to consider to write (n= 583) in the Cronbach's alpha column; and "(n= 127)" in the ICC column.

Round 2

Reviewer 1 Report

The authors improved the manuscript according with the recommendations.